# Recovery from Anesthesia after Robotic-Assisted Radical Cystectomy: Two Different Reversals of Neuromuscular Blockade

**DOI:** 10.3390/jcm8111774

**Published:** 2019-10-24

**Authors:** Claudia Claroni, Marco Covotta, Giulia Torregiani, Maria Elena Marcelli, Gabriele Tuderti, Giuseppe Simone, Alessandra Scotto di Uccio, Antonio Zinilli, Ester Forastiere

**Affiliations:** 1Department of Anaesthesiology, IRCCS Regina Elena National Cancer Institute, 00144 Rome, Italy; marco.covotta@gmail.com (M.C.); giulia.torregiani@gmail.com (G.T.); ester.forastiere@ifo.gov.it (E.F.); 2Department of Urology, IRCCS Regina Elena National Cancer Institute, 00144 Rome, Italy; gabriele.tuderti@gmail.com (G.T.); puldet@gmail.com (G.S.); 3School of Medicine, University Hospital Center “Tor Vergata”, 00133 Rome, Italy; allascotto@gmail.com; 4IRCrES, Research Institute on Sustainable Economic Growth of the National Research Council of Italy, 00185 Rome, Italy; antonio.zinilli@ircres.cnr.it

**Keywords:** anesthesia recovery periods, bladder cancer, cognitive impairment, gamma-cyclodextrins, neuromuscular blockade, robotic radical cystectomy

## Abstract

During robot-assisted radical cystectomy (RARC), specific surgical conditions (a steep Trendelenburg position, prolonged pneumoperitoneum, effective myoresolution until the final stages of surgery) can seriously impair the outcomes. The aim of the study was to evaluate the incidence of postoperative nausea and vomiting (PONV) and ileus and the quality of cognitive function at the awakening in two groups of patients undergoing different reversals. In this randomized trial, patients that were American Society of Anesthesiologists physical status (ASA) ≤III candidates for RARC for bladder cancer were randomized into two groups: In the sugammadex (S) group, patients received 2 mg/kg of sugammadex as reversal of neuromuscolar blockade; in the neostigmine (N) group, antagonization was obtained with neostigmine 0.04 mg/kg + atropine 0.02 mg/kg. PONV was evaluated at 30 min, 6 and 24 h after anesthesia. Postoperative cognitive functions and time to resumption of intestinal transit were also investigated. A total of 109 patients were analyzed (54 in the S group and 55 in the N group). The incidence of early PONV was lower in the S group but not statistically significant (S group 25.9% vs. N group 29%; *p* = 0.711). The Mini-Mental State test mean value was higher in the S group vs. the N group (1 h after surgery: 29.3 (29; 30) vs. 27.6 (27; 30), *p* = 0.007; 4 h after surgery: 29.5 (30; 30) vs. 28.4 (28; 30), *p* = 0.05). We did not observe a significant decrease of the PONV after sugammadex administration versus neostigmine use. The Mini-Mental State test mean value was greater in the S group.

## 1. Introduction

The diffusion of robot-assisted laparoscopic techniques has made it possible to perform surgical procedures with greater precision, and has reduced the need for transfusions, postoperative complications and hospitalization time [1]. In particular, robot-assisted radical cystectomy (RARC) has rapidly spread as the gold standard in the treatment of urothelial tumors, becoming a credible alternative to open cystectomy which is burdened by a high rate of complications [2].

Due to the particular surgical conditions and because of its recent application, there still are many anesthetic implications that must be examined thoroughly—patients have to satisfy specific clinical requirements, identified through careful anesthesiologic assessments [3].

During RARC, the anesthesiologist must be prepared to manage any hemodynamic, cerebrovascular and respiratory changes resulting from the surgical conditions that the robotic procedure requires, such as the prolonged use of pneumoperitoneum, the steep Trendelenburg position in which the patient is placed, and the lengthening of surgical times [4]. In addition, an effective myoresolution until the final stages of surgery is necessary to establish ideal surgical conditions [5] and the factors that can impair the quality and time of awakening [6]. To overcome this effect, a reversal of neuromuscular blockade (NMB) is routinely used in our clinical practice. 

Currently, the effectiveness of the rocuronium/sugammadex combination for the reversal of the NMB has been widely demonstrated in terms of time and quality of neuromuscular and respiratory functions [7,8].

Neostigmine has been associated with an increased incidence of postoperative nausea and vomiting (PONV), although there is no definitive agreement on the need to avoid its use to reduce the incidence of PONV [9]. On the other hand, neostigmine has an important muscarinic effect on gastrointestinal (GI) receptors, and, by increasing the availability of acetylcholine, increases the GI motility.

In our study, we investigated if the use of a different kind of NMB reversal can influence the early postoperative period after a prolonged major surgery, such as RARC, affected by alterations on mechanical ventilation, cerebral perfusion, and vascular resistances [10]. Our aim is particularly focused on PONV and ileus, with attention to the recent collective effort to build an enhanced recovery after surgery (ERAS) path applicable specifically in the interventions of RARC [11].

The hypothesis is that the continuous infusion of rocuronium followed by sugammadex administration as NMB reversal in patients undergoing robotic radical cystectomy can improve the quality of awakening in terms of postoperative outcomes and cognitive function, compared to use of neostigmine as reversal. 

The primary end point was to compare the incidence of PONV. Secondary end points were postoperative cognitive functions and time to resumption of intestinal transit (ROI).

## 2. Experimental Section

A mono-center prospective, two-arm parallel, randomized trial was conducted at the IRCCS Regina Elena National Cancer Institute. The study was approved by the Central Ethics Committee Lazio1, in May 2017, with Protocol n. CE/2288/17, and registered with ClinicalTrial.gov identifier NCT03144453. The clinical investigation was conducted according to the principles expressed in the Declaration of Helsinki.

### 2.1. Patients and Procedures

American Society of Anesthesiologists physical status (ASA) ≤III patients, candidates of RARC for bladder cancer, were enrolled after having given written informed consent. The exclusion criteria were age <18 years, inability to provide informed consent, BMI >30, and a history of cerebrovascular diseases.

Patients were randomly divided into two treatment groups by an operator who is not directly involved in the study using a specific dedicated software, developed in-house by a GW Basic (Microsoft Corporation, USA) programmer, which generates an assignment code verified immediately before arrival in the operating room. Surgeons were blinded to the intervention and blinded observers recorded the outcome.

In both groups, all patients were premedicated with midazolam 0.02 mg/kg and received dexamethasone 8 mg for anti-emesis. General anesthesia was induced with fentanyl 3–5 g/kg, propofol 2 mg/kg and a bolus of rocuronium 0.7 mg/kg was administered. After tracheal intubation, anesthesia was maintained with a mixture of sevoflurane/oxygen/air, adjusted to provide an end-tidal sevoflurane of 1.5–2 vol.%, remifentanil was adapted according to a target-controlled infusion (TCI) range of 2–4 ng/mL. Curarization started with rocuronium 5 g/kg/min and was set to maintain the post-tetanic count between 1 and 2. At the end of surgery, after skin closure, neuromuscular function was allowed to recover spontaneously and, at reappearance of the second twitch (T2), patients received a NMB reversal.

In the sugammadex group (S group), at T2 reappearance, patients received 2 mg/kg of sugammadex. 

In the neostigmine group (N group), at T2 reappearance, antagonization was obtained with the standard NMB reversal agent: neostigmine 0.04 mg/kg and atropine 0.02 mg/kg to block the peripheral muscarinic side-effects of neostigmine.

All patients were extubated when the train-of-four (TOF) ratio was 0.9 or higher. 

Nasogastric tube was removed after surgery, before the awakening.

In both groups, fluid therapy regimen was mainly restrictive, with a basal infusion of crystalloid variable from 2 to 4 mL/kg/h. Mean arterial blood pressure (MAP) was regulated by titrating remifentanil and fluid administration in order to maintain target values between 65 and 95 mmHg. 

The standard monitoring for all patients consisted of continuous ECG, heart rate (HR) and MAP measurements, pulse oximetry (SpO2), inspired and expired gas, and capnometry. Neuromuscular function was measured using a TOF-Watch acceleromyograph (Organon ltd, Dublin, Ireland). After induction of general anesthesia, but before administering any NMB agent, the calibration of the acceleromyograph was performed according to the manufacturer’s guidelines. The ulnar nerve received neuromuscular stimulation via two electrodes applied to the skin of the distal underarm, to the left and to the right of the ulnar nerve. 

The surgical procedure was performed routinely following the standards of the Department of Urology at our hospital [12].

After surgery, patients requiring rescue anti-emetic therapy received ondansetron 4 mg, which was followed by metoclopramide 20 mg, if necessary. 

All patients received intravenous morphine patient-controlled analgesia using the CADD®-Solis device (Smith Medical, Kent, UK) postoperatively. Patient-controlled analgesia was set on the demand mode without a loading dose. The dose of morphine was set at 0.02 mg/kg with a time-lock interval of 15 min. 

All patients received morphine 0.07 mg/kg and 1000 mg acetaminophen at the time of surgical wound closure, followed by 1000 mg intravenous acetaminophen every 6 h for up to 5 days.

### 2.2. Measurements

Baseline data were collected, which included risk of PONV by Apfel score, neoadjuvant chemotherapy, and anxiety and depression by the Hospital Anxiety and Depression Scale.

During anesthesia, main parameters (MAP, HR, SpO2 and etCO2) and time to recovery from NMB reversal were recorded. Duration of surgery, amount of opioid consumption, comorbidities, and total amount of intensive care unit admission were also observed. 

In the postoperative period, PONV (intended as number of episodes of nausea, vomiting or bloating) was evaluated after 30 min in post-anesthesia care unit (PACU), 6 and 24 h after anesthesia. 

The assessment of consciousness at awakening and postoperative cognitive function was carried out by The Observer’s Assessment of Alertness/Sedation Scale (OASS) at 15 min, 30 min, and 1 h after anesthesia, and through the Mini Mental State test (MMSt) at 1 and 4 h after anesthesia. 

Early postoperative pulmonary failure (including bronchospasm, postoperative PaO2 <60 mmHg, a PaO2:FIO2 ratio ≥ 300 mmHg, or arterial oxyhemoglobin saturation measured with pulse oximetry <90% and requiring oxygen therapy) was noted after 24 h after anesthesia.

Time to resumption of intestinal transit, defined as time to return of peristalsis and time to first passage of flatus, antiemetics, and morphine consumption were recorded. Nurses detected peristalsis and gastrointestinal symptoms every 2 h and patients were asked to warn staff of the perception of bowel activity. 

### 2.3. Statistical Analysis

The primary outcome was the cumulative incidence of PONV in the first 6 postoperative hours. Based on data from our department after this type of surgery using single-drug PONV prophylaxis and reversal of neuromuscular block with neostigmine, and according with previous study [13,14], we estimated that experience PONV would be 30% in neostigmine group and 8% in the sugammadex group. Based on power = 80% and *a* = 0.05, a sample size of 98 patients at least (*n* = 49 per group) was required.

For scores continuous, we used a two-sample Kolmogorov–Smirnov test, while for the ordinal categorical variable we used the Mann–Whitney U-test. *P*-values ≤ 0.05 were regarded as statistically significant. Data were analyzed using SPSS software (IBM, New York, United States).

## 3. Results

In the period between May 2017 and December 2018, a total of 109 patients were randomized: 54 patients to the S group and 55 to the N group. 

The flowchart of the patients who participated in the study is demonstrated in Figure 1. The demographics and clinical characteristics were balanced for both treatment arms and are presented in Table 1. Intraoperative and perioperative data recorded are shown in Table 2. 

Time to recovery from TOF 2 to TOF ratio >0.9 was significantly lower in the S group. The incidence of early PONV was lower in the S group but not statistically significant (*p* = 0.711). The values were similar between the two groups for the incidence of late PONV.

The mean MMSt value was significantly higher in the S group compared with the N group at 1 h after anesthesia [mean and 25–75th percentile, 29.3 (29; 30) vs. 27.6 (27; 30); *p* = 0.007] and at 4 h after anesthesia [29.5 (30; 30) vs. 28.4 (28; 30); *p* = 0.048]. Thus, S group obtained better MMSt values during all measurements of time. The mean OASS value was significantly higher in the S group compared with the N group 1 h after the end of anesthesia (median and 25–75th percentile, 5 (5; 5) vs. 5 (4; 5); *p* = 0.02), but no differences were observed in the first measurement, 30 min after the end of anesthesia (Table 3). 

In Figure 2, we can observe that the trend in both MMSt and OASS is different between the two groups. The MMSt trend remained steadily higher since the awakening, while the values of OASS in S group were significantly increased after the first postoperative hour. The incidence of postoperative pulmonary failure was similar in each group. There were no significant differences between the groups for time to resumption of intestinal transit. Postoperative ondansetron and metoclopramide were similar in each group, as well as analgesic consumption.

## 4. Discussion

In our study, we attempted to evaluate the quality of recovery from anesthesia in two groups of patients who underwent robotic-assisted laparoscopic cystectomy. Prolonged myoresolution was carried out with continuous infusion of rocuronium: In one group, NMB reversal was obtained with sugammadex and in the other group, the association neostigmine/atropine was used.

Our results show that the incidence of PONV was greater in the N group, although non-statistically significant. Even time to resumption of intestinal transit was overlapping in the two groups.

In the past, studies concerning the reduction of PONV following the use of sugammadex have had conflicting results. Inhibiting cholinesterase action causes neostigmine increases concentration of acetylcholine, the principal excitatory neurotransmitter in the GI tract. Acetylcholine acts by increasing gastric secretions and esophageal pressure and increases the risk of symptoms such as nausea and vomiting, but also allows an increase in GI motility [15]. The prevention of PONV and the rapid restoration of intestinal function are fundamental topics in the development of ERAS protocols, which have shown efficacy in reducing complications and improving outcomes in many surgeries [16]. Nowadays, there are no definitive protocols specific to robotic surgery, and protocols applied in colorectal surgery are often used for cystectomy [10]. 

Our results agree with those of Peach et al. [17], which, in a large clinical trial of 304 women, did not find a lower incidence of PONV with the use of sugammadex compared with neostigmine. In contrast, Yağan et al. [13] found that the use of sugammadex had lower incidences of PONV in the first postoperative hour and less anti-emetic use at 24 h. In addition, in the study by Koyuncu et al. [18], sugammadex reduces PONV compared with neostigmine and atropine, but only slightly and transiently. While in the Yağan study [13], the population had undergone various types of surgery (more than half underwent head and neck surgery) and in the Koyuncu study [18] patient were candidates for extremity surgery, in the Peach study [17] patients underwent laparoscopic surgery, which, as in robotic cystectomy, involves a certain degree of postoperative ileus, a physiological arrest of GI transit in response to surgical stress and intestinal manipulation. Neostigmine can increase motility only if acetylcholine release and smooth muscle function are relatively preserved, while postoperative ileus induces the activation of presynaptic noradrenergic receptors and impairs the functionality of the enteric nervous system and the sympathetic nerves [19,20]. This could have determined the absence of the expected effects on intestinal and gastric motility.

Moreover, in the study by Yağan, neostigmine doses were higher than those used in our study [13], and the correlation between the neostigmine dose and PONV is now considered a key factor to control the symptoms [9]. 

Two scales were employed as awakening quality indicators: MMSt and OASS. The MMSt was considered to assess cognitive impairment because it is a rapid and simple to perform test that provides accurate measurements of cognitive status both in subjects with normal functions and in subjects with cognitive alterations [21], and its use to assess subtle changes in cognitive function after anesthesia is often reported [22].

Our results have unexpectedly shown a significant increase of the average value of MMSt in the considered time frames. The OASS mean value also significantly increased in the S group until 1 h after surgery. 

The reversal action of sugammadex is based on the structure of cyclodextrins, consisting of a lipophilic central cavity able to encapsulate the steroid rings of the rocuronium molecule, forming an inactive complex that is no longer able to interact with the neuromuscular junction [23]. Based on its structural characteristics, the fact that the sugammadex molecule or the sugammadex/rocuronium complex could interact in any way with the anesthetic drugs or with the cholinergic system was excluded [24]. The apparent rapid awakening at a cognitive level, that some other authors and we have detected [25,26], could be explained in the light of the so-called Afferentation Theory [27], for which general activation of muscle receptors can induce a massive cerebral stimulation of the monoaminergic wakefulness centers. It is also known as the Spindle Theory [23], since it has been postulated that tension and stretch receptors in muscle spindles may be the terminations that transmit static and dynamic variations to the encephalon, acting on various cortical and mesencephalic areas. However, some studies have not been able to demonstrate changes in the depth of anesthesia after sugammadex administration [28], thus the results of studies regarding sugammadex’s impact on recovery from general anesthesia remain conflicting and insufficient [29]. 

In the past, many studies demonstrated an existing relationship between the structure of cyclodextrins and neuroprotection: statins and cyclodextrins, influence the transmission of neural signals, interfering with the production of inflammatory molecules [30]. Ultimately, one could speculate that sugammadex gives an additional effect by interacting with the lipid molecules of the neuronal membrane, reducing exocytosis. This protective effect could be more readily detectable in a surgery, such as robotic cystectomy, which requires more than 2 h in steep Trendelenburg and alterations of cerebrovascular circulation due to prolonged pneumoperitoneum [10]. In the future, it could be interesting to analyze if the use of sugammadex can be optimized, employing it in elderly populations or in surgeries that require high abdominal pressure or extreme conditions.

A limit of our study mainly regards the same limitations related to the neurophysiological tests administered in the postoperative period. These tests may be subject to the learning effect bias and to the variability in the sessions following the preoperative one, considered baseline, and from one session to another. We tried to minimize this variability by administering the test in the same environment, with no external distractions, and patients who needed extra doses of opioids for pain were excluded.

## 5. Conclusions

In conclusion, our results were not able to demonstrate a significant decrease of the PONV or a more rapid ROI after sugammadex administration versus neostigmine use. We observed a significant increase in MMSt values, suggesting improved quality of awakening with the use of sugammadex in patients undergoing robotic radical cystectomy. Regarding OASS observations in both groups, we obtained higher values in the group receiving sugammadex. Further studies on elderly populations and different types of surgery will be needed in the future, especially with the aim to provide a comprehensive ERAS pathway for cystectomy based on the available evidence.

## Figures and Tables

**Figure 1 jcm-08-01774-f001:**
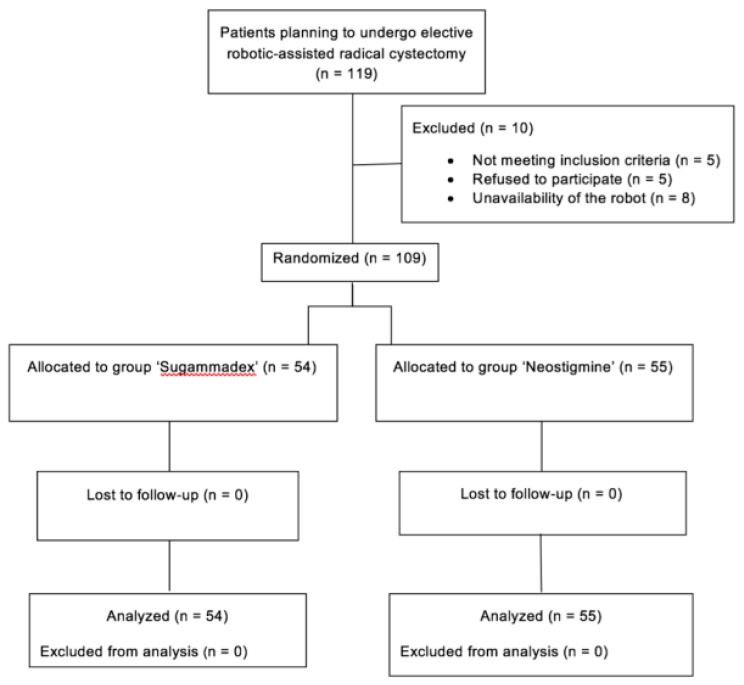
Patient disposition.

**Figure 2 jcm-08-01774-f002:**
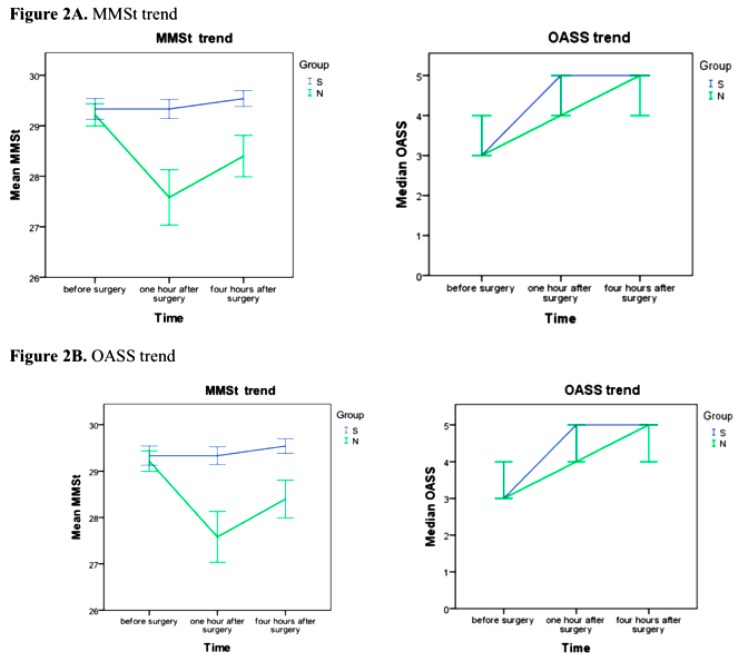
(**A**) MMSt and (**B**) OASS trend (error bars: 95% CI). Blue line: S group. Green line: N group.

**Table 1 jcm-08-01774-t001:** Demographic and clinical characteristics.

	S Group (*n* = 54)	N Group (*n* = 55)
Age (years), mean (SD)	62.8 (8.9)	60.2 (9.4)
BMI (kg/m^2^), mean (SD)	26.3 (3.5)	26.2 (4)
Gender (n), male/female	42/12	40/14
ASA status (n): I/II/III	5/40/9	9/41/5
Apfel risk score (n): I/II/III/IV	20/30/4/0	19/32/4/0
Comorbidities, n (%)		
Hypertension	18 (33.3)	11 (20)
Dysthyroidism	3 (5.5)	4 (7.2)
Previous MI	5 (9.2)	2 (3.6)
Diabetes	6 (11.1)	3 (5.4)
COPD	3 (5.5)	2 (3.6)
Neoadiuvant chemotherapy, *n* (%)	14 (25.9)	11 (20)
Tumor stage (pT), *n* (%)		
Tis	8 (14.8)	7 (12.7)
Ta	3 (5.5)	3 (5.4)
T1	7 (13)	8 (14.8)
T2	13 (24)	14 (25.4)
T3	17 (31.4)	16 (29)
T4	6 (11.1)	7 (12.7)
HADS > 8, *n* (%)	25 (46.2)	27 (49)

BMI: body mass index; ASA: American Society of Anesthesiologists; COPD: chronic obstructive pulmonary disease; HADS: Hospital Anxiety and Depression Scale.

**Table 2 jcm-08-01774-t002:** Intraoperative and perioperative variables.

	S Group (*n* = 54)	N Group (*n* = 55)	*p*-Value
EtCO_2_ (mmHg)	28.9 (3)	28.6 (3.4)	0.603
SpO_2_ (%)	98.6 (1.3)	98.5 (1.5)	0.821
HR (bpm)	68.3 (15.1)	68.9 (13.9)	0.622
MAP (mmHg)	87 (15.3)	88.2 (15.5)	0.854
Estimated blood loss (mL)	209 (31)	218 (37)	0.200
Surgery time (min)	340.7(80)	326.7 (81.9)	0.437
Anesthesia time (min)	378 (83)	361 (81)	0.526
Recovery time from TOF 2 to TOF Ratio > 0.9 (min)	3.2 (1)	8 (2.8)	<0.001 *
Early PONV 0–6 h, *n* (%)			
Cumulative incidence	14 (25.9)	16 (29)	0.711
Nausea	10 (18.5)	9 (16.3)	0.767
Vomiting	4 (7.4)	5 (9)	0.750
Late PONV 6–24 h, *n* (%)			
Cumulative incidence	10 (18.5)	11 (20)	0.845
Nausea	7 (13)	8 (14.5)	0.810
Vomiting	3 (5.5)	3 (5.4)	0.982
Antiemetics consumption (mg)			
Ondansetron	2.6 (3)	3.8 (4.4)	0.105
Metoclopramide	3.7 (4.9)	4.7 (5.7)	0.358
Morphine consumption (mg)			
0–6 h	3 (2.4)	3.7 (2.6)	0.154
0–24 h	6.2 (3)	5.5 (2.8)	0.177
Early postoperative pulmonary failure, *n* (%)	3 (5.5)	4 (7.2)	0.715
Time to resumption of intestinal transit, days (IQR)	3 (3–5)	3 (3–5)	0.761
Length of stay, days (IQR)	8 (7.5–12.25)	8 (6–12)	0.682

* *p*-value < 0.05; EtCO_2_: end tidal CO_2_; SpO_2_: pulse oximetry; HR: heart rate; MAP: mean arterial pressure; TOF: train-of-four; PONV: postoperative nausea and vomiting; IQR: interquartile range.

**Table 3 jcm-08-01774-t003:** Consciousness at awakening and postoperative cognitive function.

	S Group (*n* = 54)	N Group (*n* = 55)	*p*-Value
OASS ^¢^			
15 min	3 (3; 4)	3 (3; 4)	0.16
30 min	5 (4; 5)	4 (3; 5)	0.06
60 min	5 (5; 5)	5 (4; 5)	0.023 *
MMSt ^♯^			
Preop	29.3 (30; 30)	29.2 (29; 30)	0.78
1 h	29.3 (29; 30)	27.6 (27; 30)	0.007 *
4 h	29.5 (30; 30)	28.4 (28; 30)	0.048 *

^♯^ Data expressed as mean (IQR); ^¢^ Data expressed as median (IQR); *p*-value: two-sample Kolmogorov–Smirnov Test; * *p*-value < 0.05; OASS: Observer’s Assessment of Alertness/Sedation Scale; MMSt: Mini Mental State test: IQR: interquartile range.

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
