# Peer review of "Recovery from Anesthesia after Robotic-Assisted Radical Cystectomy: Two Different Reversals of Neuromuscular Blockade"

_jcm, 2019, doi:10.3390/jcm8111774_

Round 1
Reviewer 1 Report
This is an interesting study. The authors would be advised to present their data more clearly: Figure 2 depicts the observations in regards to MMSt and OASS but the figure legend fails to explain the graphs. Similarly, the result section only peripherally touches on a description of these results which are among the key findings of the study.
Otherwise the design is sound and well-executed and the Discusion is well-written warranting publication in JCM.
Author Response
This is an interesting study.
The authors would be advised to present their data more clearly: Figure 2 depicts the observations in regards to MMSt and OASS but the figure legend fails to explain the graphs. Similarly, the result section only peripherally touches on a description of these results which are among the key findings of the study.
Thank you for this suggestion. We added observations in Results section and clarified the figure 2 legend.
Reviewer 2 Report
In their randomized trial, the authors aimed at comparing the postoperative nausea and vomiting (PONV) in patients receiving sugammadex (S) versus neostigmine (N) after robot-assisted radical cystectomy (RARC) for bladder cancer (BC). A total of 109 patients were enrolled. The incidence of early PONV was lower in the S group but not statistically significant (S group 25.9% vs N group 29%; p=0.711). On the other hand, The Mini-Mental State test mean value was higher in the S group. Overall, it is a well conducted study with precise methodology.
The following comments are made:
Even if sample size analysis was correctly assessed in the trial protocol, the pre-assessed expected difference of 22% of PONV between these two groups may represent a flaw of the study not because the sample was by itself incorrect, but because the expected difference was overestimated. Could the authors provide further evidence from previous literature supporting the expected difference in PONV defined in the trial protocol?
Do the authors have data on adverse event rates between the two groups?
Author Response
Even if sample size analysis was correctly assessed in the trial protocol, the pre-assessed expected difference of 22% of PONV between these two groups may represent a flaw of the study not because the sample was by itself incorrect, but because the expected difference was overestimated. Could the authors provide further evidence from previous literature supporting the expected difference in PONV defined in the trial protocol?
Thanks for this interesting observation. As explained in the text we assessed the expected difference of 20% of PONV between the groups based on data from our previous experience in the same operative setting; however, according to your advice, we added in the text some studies supporting this expected difference [ref. 13,14]